# Analysis of Transcriptional Changes in Different *Brassica napus* Synthetic Allopolyploids

**DOI:** 10.3390/genes12010082

**Published:** 2021-01-11

**Authors:** Yunxiao Wei, Guoliang Li, Shujiang Zhang, Shifan Zhang, Hui Zhang, Rifei Sun, Rui Zhang, Fei Li

**Affiliations:** 1Institute of Vegetables and Flowers, Chinese Academy of Agricultural Sciences, Key Laboratory of Biology and Genetic Improvement of Horticultural Crops of the Ministry of Agriculture, Beijing 100081, China; weiyunxiao@caas.cn (Y.W.); liguoliang@caas.cn (G.L.); zhangshujiang@caas.cn (S.Z.); zhangshifan@caas.cn (S.Z.); zhanghui05@caas.cn (H.Z.); sunrifei@caas.cn (R.S.); 2Biotechnology Research Institute, Chinese Academy of Agricultural Sciences, Beijing 100081, China

**Keywords:** *Brassica napus*, allopolyploid, transcriptome, expression level dominance, trait separation

## Abstract

Allopolyploidy is an evolutionary and mechanistically intriguing process involving the reconciliation of two or more sets of diverged genomes and regulatory interactions, resulting in new phenotypes. In this study, we explored the gene expression patterns of eight F2 synthetic *Brassica napus* using RNA sequencing. We found that *B. napus* allopolyploid formation was accompanied by extensive changes in gene expression. A comparison between F2 and the parent shows a certain proportion of differentially expressed genes (DEG) and activation\silent gene, and the two genomes (female parent (AA)\male parent (CC) genomes) showed significant differences in response to whole-genome duplication (WGD); non-additively expressed genes represented a small portion, while Gene Ontology (GO) enrichment analysis showed that it played an important role in responding to WGD. Besides, genome-wide expression level dominance (ELD) was biased toward the AA genome, and the parental expression pattern of most genes showed a high degree of conservation. Moreover, gene expression showed differences among eight individuals and was consistent with the results of a cluster analysis of traits. Furthermore, the differential expression of waxy synthetic pathways and flowering pathway genes could explain the performance of traits. Collectively, gene expression of the newly formed allopolyploid changed dramatically, and this was different among the selfing offspring, which could be a prominent cause of the trait separation. Our data provide novel insights into the relationship between the expression of differentially expressed genes and trait segregation and provide clues into the evolution of allopolyploids.

## 1. Introduction

Polyploidy, or whole-genome duplication (WGD), is prevalent in nature and is particularly common in angiosperms, increasing biodiversity and providing new genetic material for evolution [1,2,3]. Synthetic polyploidy is often associated with novel and presumably advantageous ecological attributes such as range expansion [4], novel secondary chemistry, morphology [5], and increased pathogen resistance [6]. Previous studies have investigated synthetic allopolyploids and show that various genetic [7,8] and epigenetic [9,10,11] changes, as well as alterations in gene expression levels [12,13,14] occur at the initial stage of allopolyploidization. At the genetic level, loss of parental and/or appearance of novel sequences are common events at the initial stage of allopolyploidization. Non-homologous chromosome exchanges occur in synthetic *B. napus*, resulting in the addition and/or deletion of sequences [15]. At the epigenetic level, changes in small RNA and DNA methylation patterns occur at the initial stage of allopolyploidization. Shen et al. have reported higher siRNA and DNA methylation levels in F1 hybrids [16]. The role of heredity and epigenetics leads to changes in gene expression, which in turn leads to novel phenotypes [17].

The development of RNA-seq and the release of numerous plant genome sequences have allowed extensive studies on expression profiling of various allopolyploids, such as *Arabidopsis*, cotton, wheat, and *Brassica*. Fortunately, sequencing and assembly work on the female parent (AA) genome [18], male parent (CC) genome [19], and F1 allopolyploid (AACC) genome [20] have been completed. In the synthetic allopolyploid *Arabidopsis suecica* leaf transcriptome, the expression of numerous homologous genes was biased towards the parent *Arabidopsis arenosa* [21]. However, in synthetic allopolyploid and nature cotton, the number of partially homologous pairs favoring the AA genome expression and biased D genome expression is basically the same [14]. In addition, Li et al. [22] found that nonadditive expressed genes were rare but relevant to growth vigor in synthetic wheat. Furthermore, Gaeta et al. [15] indicate that exchanges among homoeologous chromosomes are a major mechanism creating novel allele combinations and phenotypic variation in newly formed *B. napus* polyploids. Doyle et al. put forward that novel expression diversity and genetic alterations generate evolutionarily novel phenotypes [23].

*B. napus* is an important oil crop and is a hybrid that was generated from a cross between *Brassica rapa* and *Brassica oleracea* approximately between 6800 and 12,500 years ago [20]. *B. napus* germplasm resources are scarce due to its relatively short formation time [24]. However, *B. rapa* and *B. oleracea* have rich germplasm resources. Breeders generally use distant crosses to broaden their resources [25,26]. However, breeders are concerned that the self-progeny of synthetic allopolyploids results in trait separation [24]. Breeders thus are unable to fully utilize these synthetic resources. Therefore, research on the mechanism underlying trait separation is warranted [26].

Previous studies have explored changes in gene expression profiles in synthetic first generation (F1) allopolyploids [27,28,29]. However, these patterns in allopolyploids with different traits have not been fully examined. The present study analyzed the gene expression patterns of eight second generation (F2) synthetic allopolyploids using RNA-seq. We extend earlier findings of homoeolog expression bias and expression level dominance by parsing expression patterns in eight synthetic allopolyploids and their parents. The experiment also analyzed the relationship between gene expression difference and trait separation in eight plants, taking the waxy and flowering traits as an example to show.

## 2. Materials and Methods

### 2.1. Plant Materials

For this study, we used ten accessions, including the female parent Cai-Xin, male parent Chinese kale, and eight F2 synthetic allopolyploids (Figure 1). First, by embryo rescuing, F1 haploid (AC) hybridization between Cai-Xin and Chinese kale was performed. Then, F1 allopolyploids (AACC) were obtained by colchicine doubling [30]. Seeds were collected by F1 (AACC) budding self-pollination. The eight F2 plants and the parents were planted in the greenhouse of the Chinese Academy of Agricultural Sciences Institute of Vegetables and Flowers (Beijing, China). We investigated the field traits during the flowering period: flower time, stalk diameter, plant height, fourth leaf length, fourth leaf width, petiole long, flower size, bud size, leaf wings, leaf color, and leaf shrinkage. 

### 2.2. RNA Extraction and Library Preparation

Young leaves next to bud (5 cm in length) were collected. After the leaves were taken, they were frozen in liquid nitrogen and stored at −80 °C until extraction. RNA was extracted from three biological replicates using TRIzol reagent (Invitrogen, Life Technologies, Beijing, China) following standard protocols. The quality and quantity of the extracted RNA were assessed using the NanoPhotometer^®^ spectrophotometer (IMPLEN, San Diego, CA, USA). RNA-Seq libraries were generated using the NEBNext^®^ Ultra™ RNA Library Prep Kit for Illumina^®^ (NEB, Boston, MA, USA). OligodT magnetic beads were used to enrich mRNA. The library preparations were sequenced on an Illumina Hiseq platform (Beijing, China) and 150 bp paired-end reads were generated.

### 2.3. Read Filtering, Mapping, and Analysis of Differentially Expressed Genes

After high-throughput sequencing, the raw data of 30 samples that contained adapters were trimmed, and low-quality reads were filtered. Then, the clean and properly paired reads were aligned to the *B. rapa* [18] and the *B. oleracea* [19] reference genomes using HISAT2, and the number of reads that could be mapped to genes was determined. Then, the fragments per kilobase of exon per million reads mapped (FPKM) of each gene was calculated based on the length of the gene and the number of reads that were mapped to this gene. Differential expression analysis of two conditions (three biological replicates per condition) was performed using the DESeq R package [31]. The resulting *p*-values were adjusted using Benjamini and Hochberg’s approach to control false discovery rates [32]. Genes with an adjusted *p*-value < 0.05 identified by DESeq were designated as differentially expressed.

### 2.4. Annotation of Orthologous Genomic Regions

To identify orthologous genomic regions between the two reference genomes, the AA genome gene model was aligned against the genomic sequence of the C genome sequence using BLAST with a cutoff E-value of e-10 [33]. The annotated A genome genes in the C genome that were located in orthologous genomic regions and showed sequence identity of >90% with the C gene model were considered homologous genes between A and C.

### 2.5. Analyses of Expression Level Dominance and Homoeolog Expression Bias

To assess the direction of expression level dominance, we compared expression levels between synthetic allopolyploids and their parents. The genes were divided into 12 expression patterns with classifications based on differential expression ((*p* < 0.05) or not (*p* ≥ 0.05)) [27]. To analyze the biased expression of homologous genes, we selected the homoeologous gene pairs between the AA and CC genomes to determine the direction of the homoeolog bias. In addition, a comparison between their progenitors was performed to estimate whether the gene expression patterns would be inherited to offspring [14,27].

### 2.6. Gene Ontology (GO) Enrichment Analysis

Gene Ontology (GO) classification of each gene model was performed using BLAST. GO enrichment analysis was implemented by the GOseq R package, in which gene length bias was corrected. GO terms or slims with corrected *p* < 0.05 were considered significantly enriched.

## 3. Results

### 3.1. Transcriptome Sequencing and Read Mapping

Three biological replicates of each genotype were sampled. In total, 30 RNA libraries were subjected to paired-end RNA sequencing, and 1.55 billion clean reads were obtained with an average of 51.6 million reads (7.7 Gb) in each sample (Appendix A). On average, 78.04% and 68.93% of the reads from the flowering Chinese cabbage and Chinese kale samples were uniquely mapped to the *B. rapa* and *B. oleracea* genome sequences, respectively. Regarding the resynthesized *B. napus* samples, 63.7% and 54.3% (an average of eight plants) of the reads were uniquely mapped to integrated genomes of the *B. rapa* and *B. oleracea* (Appendix A). The gene expression correlations between each pair of biological replicates were strong, with most Pearson correlation coefficients (R) > 0.81 (Appendix A). These results indicate that the sequencing data of the biological replicates were of high quality.

### 3.2. Differential, Non-Additive Gene Expressions in Synthetic Allopolyploids

We constructed and sequenced 30 RNA-seq libraries, including 24 synthetic allopolyploids samples and six parental samples. First, we compared the gene expression profiles between the female and male parents by homologous genes (Figure 2a). A total of 8123 genes (31%) were differentially expressed, of which 4381 (17%) were upregulated in CC, and 3742 (14%) were upregulated in AA. Comparisons between synthetic allopolyploids and parental diploids also showed a high fraction of differentially expressed genes, with equivalent proportions 23% vs. 12% in AACC1, 27% vs. 7% in AACC2, 25% vs. 9% in AACC3, 23% vs. 7% in AACC4, 24% vs. 8% in AACC5, 28% vs. 10% in AACC6, 24% vs. 8% in AACC7, and 15% vs. 5% in AACC8 (Figure 2a). The number of differentially expressed genes (DEGs) compared with the male parent was significantly higher than the number of genes when compared with the female parent (*p* < 0.05, *t*-test). Moreover, alignment to the CC genome indicates no significant bias between the up and downregulated genes in the synthetic allopolyploid relative to the parental diploids (*p* > 0.05, *t*-test), whereas alignment to the AA genome showed a higher proportion of upregulated than downregulated genes in synthetic allopolyploids relative to the parental diploids (*p* < 0.05, *t*-test). The result showed that the two genomes (AA\CC genomes) showed significant differences in response to WGD, the CC genome showed a bigger change than AA genome. In addition, GO enrichment analysis revealed differences between different genomes and individual pants (Appendix A). The GO enrichment items for two genomes and each individual were different, indicating that the expression of differential genes laid the groundwork for the field difference performance of plants.

We then compared the gene expression levels of the eight plants to mid-parent expression values (MPV) to assess non-additive expression (Figure 2a). The majority of genes exhibited additivity, and non-additively expressed genes only represented a small portion of the expressed genes (16% in AACC1, 15% in AACC2, 16% in AACC3, 11% in AACC4, 12% in AACC5, 18% in AACC6, 12% in AACC7, and 5% in AACC8) in the synthetic hybrids. The result showed that non-additive expressed genes account for a small number., The proportion of different individual plants was different, and the difference was large (5%–16%). In addition, the non-additive expression genes had the same different genes among eight single plants (Figure 2b). The intersection of different plants was small, and the proportion of the set was large. There were many genes that are specifically expressed by each plant. AACC6 had the highest number of gene-specific expressions. AACC1 followed by AACC6. We speculated that these specifically expressed genes play an important role in field difference performance. Furthermore, GO enrichment analysis found that most of them are enriched in the process of defense, anti-adverse reaction, and response to hormones (Figure 2c). The hypothesis of “non-additive gene expression and multiple molecular mechanisms promoting heterosis” was matched, thereby demonstrating a hybridization advantage at the gene expression level [26].

### 3.3. Expression Level Dominance in Different Synthetic Allopolyploids

To detect additivity, transgressive expression, and expression level dominance, the genes in synthetic allopolyploids were binned into 12 categories (Figure 3a). The additivity gene in the eight synthetic allopolyploids are 13.4%, 11.9%, 12.7%, 13.5%, 13.1%, 11.7%, 13.2%, 18.0% ((I + XII)/25,883). In general, a higher number of transgressive upregulated (categories V, VI, and VIII) genes in the eight synthetic allopolyploids was observed relative to the number of downregulated (categories III, VII, and X) genes in the synthetic allopolyploids (14.8% vs. 3.9%). We examined all four progenies (II, IV, IX, and XI) for evidence of expression level dominance. The expression levels of genes belonging to categories IV, and IX were statistically equivalent to that of the A genome parent, and genes of categories II and XI were the same as that of the C genome parent. The percentage toward the A genome parent and C genome parent in the eight synthetic allopolyploids were 19.3% vs. 8.9%, 26.5% vs. 4.1%, 22.1% vs. 5.6%, 22.1% vs. 5.0%, 22.7% vs. 5.3%, 25.2% vs. 6.0%, 22.3% vs. 5.2%, and 13.9% vs. 4.1%. respectively. Thus, an average of 16.3% more gene pairs (2819 (21.8%) vs. 715 (5.5%)) exhibited expression level dominance toward the A parent than the C parent (Figure 3b). In the allotetraploid of *Arabidopsis thaliana*, the cis-acting elements and trans-acting factors carried by different parents are different, and their regulatory effects on homologous genes were also different. This difference in cis-trans regulation may lead to genomic biases in gene expression [34]. 

Furthermore, we analyzed the difference set, intersection, and set of BIA_A (IV + IX) (Gene biased to the expression of the A genome), BIA_C (II + XI) (Gene biased to the expression of the C genome), high_parents (II + IV) (Gene higher than the expression of the parent), down_parents (IX + XI) (Gene lower than the expression of the parent), transgressive-up regulation (V + VI + VIII) transgressive-down regulation (III + VII + X). We found that the intersection of different plants was small, and the proportion of the set was large. Each plant had many genes that were expressed uniquely. In addition, the difference sets of AACC1 accounted for the highest, followed by AACC6 (Figure 4a). Namely, the number of genes expressed uniquely was more than that of other plants. Interestingly, we use trait data to grade and do cluster analysis. Field traits included flowering time, stalk diameter, plant height, fourth leaf length, fourth leaf width, petiole long, flower size, bud size, leaf wings, leaf color, and leaf shrinkage (Appendix A), which were surveyed at the time of taking samples. Lastly, the cluster map displayed that AACC1, AACC6 were farther away from other plants (Figure 4b). The results showed that the gene expression of the six patterns in the eight individuals was consistent with the trait cluster analysis. The genes with different expression patterns of eight plants had large differences, which lays the molecular basis for the difference of individual plant characters.

Besides, the gene expression of transgressive-upregulation and -downregulation were analyzed for GO enrichment. It was found that the transgressive-upregulated genes were mainly enriched in the processes of stress resistance, enzyme activity, and carbohydrate metabolism (Figure 4d), while transgressive-downregulated genes were mainly enriched in DNA assembly, nuclear small body, chromatin assembly, protein complex assembly, and other processes (Figure 4c). Transgressively expressed genes play an important role in responding to genome shock [35], and the performance of hybridization and genomic instability are associated with these genes. The results showed that the transgressive-upregulated genes might play a role in heterosis, and transgressive-downregulated genes might explain genetic changes caused by the combination of two genomes.

### 3.4. Homoeolog Expression Bias in Synthetic Allopolyploids

To estimate the extent of homoeolog expression bias, the expression levels of homoeologous gene pairs between the parental diploids and synthetic allopolyploids were compared (Figure 5a). The expression patterns of the parental diploids showed high conservation in the synthetic allopolyploids. The gene of the parental condition in the eight synthetic allopolyploids respectively were 71.3%, 72.3%, 73.4%, 72.0%, 73.4%, 70.2%, 72.9%, and 73.4% (Figure 5b). In addition, approximately 7.8%–12.4% of the gene pairs in the synthetic allopolyploids exhibiting preexisting expression bias reverted to non-differential expression. In contrast, 3.3%–12.7% gene pairs in synthetic allopolyploids exhibiting non-differential expression reverted to expression bias. Notably, the resynthesized allotetraploid showed unbalanced biased expression with a preference toward the C genome in all eight plants (A-bias vs. C-bias = 2248 (8.69%) vs. 2697 (10.44%), Appendix A). Unbalanced homoeolog expression bias in allopolyploids is commonly observed, varies in magnitude, and remains mechanistically mysterious [36]. Furthermore, AACC6 showed the highest percentage of the “parental condition”, and the lowest percentage of “no bias in hybrid”, and “novel bias in hybrid”. In contrast, AACC8 exhibited the opposite trend. The result showed that similar to the results of non-additive gene expression, we also observed differences among the eight plants.

### 3.5. Novel Expression/Silencing Analysis in Synthetic Allopolyploids

Novel expression was inferred when both parental species had no reads for a gene, yet synthetic allopolyploids displayed more than ten read counts per gene per million reads in all three biological replicates. If both parental species had more than ten read counts per homoeolog per million reads, but synthetic allopolyploids had zero counts for the same homoeolog, then the gene was considered silenced [37]. Figure 5c shows that the number of genes in NOVEL_AA and NOVEL_CC was similar, while the number of genes in SILENCE_AA and SILENCE_CC was significantly different. The experiment again confirms that the two genomes were different in response to genome shock. As with the results of the DEG analysis, the CC genome showed a bigger change than the AA genome. In addition, the difference set, intersection, and set of activating and silencing genes in eight plants were analyzed (Figure 5d). The difference set was small, and the number of sets was large; the intersection was also relatively large, indicating that the gene activation and silent expression were similar among the individual plants. Different from the results of non-additive expressed genes: eight plants retained many genes that were specifically expressed by themselves. The results showed that the silenced gene and the novel expressed gene were not significantly different in eight plants.

### 3.6. Analysis of the Genes Related to Wax Synthesis and Flowering Pathway in the Eight Allopolyploids

Based on the genes in the waxy synthetic pathway in Arabidopsis, we found homologous genes in the AA and CC genomes and observed differential expression of these homologous genes in eight plants (Figure 6a,c). In Figure 6c, the straight line represents the differential expression of the paternal homologous gene, and the dotted line represents the expression of the maternal homologous gene. Different genomes had inconsistent responses to genome shock. Furthermore, eight plants performed differently in the field: the leaves showed varying degrees of waxiness (Appendix A). It was speculated that some homologous genes related to the waxy pathway were different in the eight individuals laid a molecular basis for the differences in wax accumulation on the surface of different leaf plants. Interestingly, the *CER1* homologous gene in *B. rapa* was upregulated in AACC4 and AACC5, while it showed no change in other plants. The *CER1* homologous gene in *B. oleracea* was down-regulated in AACC4 and AACC5. Consistent with field performance, AACC4 and AACC5 showed no wax powder (Appendix A). In addition, Liu et al. localized the waxy deletion gene of *B. oleracea*, which was a homologous gene of *CER1*. The down-regulated expression of the *CER1* gene inhibits the decarboxylation of aldehydes, resulting in the reduction of long-chain alkane synthesis [38]. Moreover, the CER1 protein could form a heterodimer with the CER3 protein to synergistically catalyze the reduction of acyl coenzyme-A to aldehydes and subsequent decarboxylation of aldehydes to alkanes [39,40,41]. It was speculated that the dosage effect of two genomic *CER1* homologous genes might affect the function of CER1 and CER3 protein complexes and further reduce the synthesis of alkanes. Eventually, AACC4 and AACC5 showed no wax. In summary, the results showed that the homologous genes of the two genomes work together, resulting in changes in traits. The *CER1* gene could play a key role in leaf wax-free performance.

Using the same method, we analyzed homologous genes in the flowering pathway in the AA and CC genomes and observed differential expression of these homologous genes in eight plants (Figure 6b). The result showed that the differential expression of homologous genes in *B. oleracea* leads to earlier flowering time. Down-regulation of the *AP2* and *AP2*-like gene leads to early flowering time, as they are transcriptional repressors of the *FT* gene. The *PHYB* gene was the same, which negatively regulates the *CO* gene, and the *CO* gene is the major regulator of the *FT* gene. However, the differential expression of homologous genes in *B. rapa* resulted in a delayed flowering time. The upregulated expression of the *PHYB* gene and the downregulated expression of the *CO* gene caused a delay in flowering time. The *FLC* gene was the same, which is a strong inhibitor of the *FT* gene. Consistent with field performance, the flowering time of eight plants was longer than that of the female parent and shorter than that of the male parent (Appendix A). In addition, we also found that AACC1 showed the largest number of differentially expressed genes. The *SVP* and *VIN3-like* genes appeared to be flowering inhibitors, while the *CSTF64* gene was a positive regulator of flowering. The *TOE3* gene is a transcriptional repressor of the *FT* gene, while the *SOC1* gene is a characteristic flower meristem gene. Differential expression of these genes resulted in delayed flowering time. Consistent with field performance, AACC1 flowering time was the latest (Appendix A). Therefore, differential gene expression in synthetic allopolyploids was the cause of trait separation.

## 4. Discussion

### 4.1. Non-Additive Expression Patterns Performance Difference in Eight Synthetic Allopolyploids

Polyploid gene expression is not the sum of the two-parental expression of the average, but there is a lot of non-additive expression changes [12,14,16,28,42,43]. Through the transcriptome study of eight synthesis allotetraploids and their parents of *B. napus*, we found that although most of the genes after the merger of the two genes showed additive expression, However, many genes still had non-additive expression, of which expression was different between hybrids and mid-parents.

The average (13.13%) proportion of non-additive expression genes is higher than that of cotton (4.9%) hybrids [14], synthetic *Arabidopsis* allotetraploids (5.2%–5.6%) [12], synthetic wheat heterologous hexaploids (0.9%) [22], and synthetic *Brassica* heterogeneous hexaploids (7.8%) [28]. Zhang et al. found that the proportion of non-additive expression genes in synthetic *B. napus* was 16%, which is similar to our results [27]. The mixed sampling method used by Zhang et al. (2016a) may have ignored interindividual differences. Our experiment conducted sampling from a single plant, which takes into account the difference. The results showed that non-additive expression significantly differed among the eight synthetic allotetraploids of *B. napus*. This also proves that the newly synthesized allopolyploid is unstable, and the expression of genes in the selfed-offspring drastically changes [44,45]. Furthermore, the results of GO enrichment analysis were consistent with the hypothesis that “non-additive gene expression and multi-molecular mechanism promote heterosis”, which also indicates that non-additive expression genes play a key role in the field performance of offspring [26].

### 4.2. Expression Level Dominance Biased to A Genome in Eight Synthetic Allopolyploids

Rapp first coined the concept of expression level dominance (ELD) [46], wherein a gene is biased toward the expression of one parental genome. This phenomenon has been observed in various heterologous polyploids such as cotton [14,47], wheat [22,42,48], and *Spartina* [13]. Zhang et al. aligned the genes of synthetic allopolyploids to the *B. napus* genome [20], using all genes for an expression level dominance analysis, which showed that allotetraploids (AACC) showed no obvious genome bias [28]. Our experiment respectively aligned the gene of eight allopolyploids to *B. rapa* [18] and *B. oleracea* [19] and used homologous genes for expression level dominance analysis. It was found that the genes of eight synthesis allotetraploids were biased towards A genome expression. The results of the research by Wu et al. were consistent with ours, and they used the same method as used here [29]. The discrepancies in the results may be due to the methodology used in each study [13]. The constituent An and Cn subgenomes are engaged in subtle structural, functional, and epigenetic cross-talk, with abundant homeologous exchanges [20]. The method of our experiment may thus accurately reflect the expression of crossbreeds. Furthermore, the subgenome differences in cis-acting elements, trans-acting factors, TE density, and methylation probably underlie subgenome dominance [34,49,50,51,52]. Therefore, the regulatory element, TE, and epigenetic differences of eight individual plants need to be further explored. 

In addition, BIA_A, BIA_C, high_parent, down_parent, transgressive-upregulation, and -downregulation, which were six combinations of 12 gene expression patterns, showed differences in eight plants, which were similar to the results of cluster analysis of traits. The results show that due to the newly synthesized heterologous diploid self-progeny presented with different gene expression, six gene expression patterns may be strongly associated with the phenomenon of trait separation. Drastic changes in gene expression lay the molecular foundation for trait separation. In addition, the results of super-parent expression GO enrichment analysis indicated that different expression pattern genes play different roles in the performance of offspring, including field trait performance and maintenance of genomic balance. The specific mechanism remains to be further explored. 

### 4.3. Homoeolog Expression Bias Performance no Difference in Eight Synthetic Allopolyploids

The biased expression of homologous genes from different heterologous polyploids has been extensively investigated [9,13,14,16,22,47]. By comparing the expression of homologous genes in parents and allopolyploids, we found that “parental condition” accounts for the largest proportion, and “novel bias in hybrid” accounts for the smallest proportion. This shows that most of the parental expression patterns are transmitted to the offspring. This phenomenon has been observed in cotton and wheat [22,26]. In addition, we found that homoeologous gene number of A-bias has no significant difference with C-bias, which is similar to the findings of Shen et al. [16] and in contrast to that of Zhang et al. [27]. These differences in results may be due to material differences and tissue-differentiated expression [13,35]. Our data supports the theory that homoeologous expression bias in synthetic allopolyploids is unbalanced. However, the mechanism of unbalanced homoeolog expression bias in allopolyploids remains mysterious [36]. The provision of transcriptome data from more heteropolyploids and tissues may be helpful in exploring this issue.

### 4.4. Differential Gene and Novel/Silenced Expression Analysis in Eight Synthetic Allopolyploids

Due to the small number of synthetic heterologous polyploids sequenced, few studies have investigated the significant differences in the differentially expressed genes between the synthetic AACC allotetraploid hybrids and the two parents. Experimental results showed that the differential gene aligned with AA was significantly higher than CC, and the upregulated expression was higher than the number of downregulated genes. In addition, this experiment also analyzed gene silencing\activation. Gene silencing\activation refers to the phenomenon that genes change from expression to non-expression due to epigenetic or genetic influences and could occur rapidly after polyploidy formation [13,35]. Because of the small number of sequencing progeny and the difference in comparison methods, the predecessors rarely compare the differences in gene expression between the two genomes [14,35]. Our results show that the number of genes showing novel expression patterns or silencing differed when aligned to the two parents. A similar finding was observed for differentially expressed genes. This indicates that there was a difference between the two genomes in response to WGD. The CC genome showed a bigger change than the AA genome. The mechanism behind this phenomenon needs to be further explored.

### 4.5. Wax Synthesis and Flowering Pathway Among the Eight Synthetic Allotetraploids

The role of genetics and epigenetics leads to changes in gene expression, which in turn leads to altered traits [17,23,53]. Gaeta et al. have suggested that exchanges between non-homologous chromosomes in artificially synthesized *B. napus* lead to trait separation [15]. In *Arabidopsis* allopolyploids, Gyoungju et al. put forward that flowering time variation is probably related to the expression diversity and/or copy number of multiple FLC loci [54]. In our experiments, it was found that the genes of the two genomes in the waxy synthetic pathway were differentially expressed in the eight individual strains, and it was found that the *CER1* gene may play a key role due to the dose-effect of homologous genes. In addition, it was found that the genes of the two genomes in the flowering pathway were significantly different in AACC1 and other individuals, explaining that AACC1 flowered at the latest. The experiment attempted to explain the phenomenon of trait segregation at the level of the trait pathway. The difference in gene expression between the two genomes was obvious. It was proved that the differential expression of genes laid the foundation for the separation of traits. 

When F1 (AACC) forms gametes via meiosis, exchanges between non-homologous chromosomes make each gamete carry different genetic information, leading to different gene expression patterns and trait separation [55]. It means that eight plants carried different genetic information, although these came from the same homozygote. Our experiment provides a new perspective on gene expression from selfing offspring of synthetic allotetraploids. In addition, the *cis*- and *trans*-acting elements are factors that lead to differences in gene expression [17]. Allopolyploids also undergo epigenetic changes such as small RNA and DNA methylation [16,22]. Thus, additional investigations of the epigenetic and genetic changes among the eight plants are warranted.

## 5. Conclusions

In this experiment, a gene expression analysis was performed on eight F2 synthetic *B. napus* and their parents. Including differential gene analysis, non-additive gene analysis, expression level dominant analysis, homology bias expression analysis, gene silencing\activation, and waxy synthesis pathway, flowering pathway gene expression analysis. The results show that the proportion of differential genes that are aligned with the AA\CC genome is 5%–12%, 15%–28%. The number of gene silencing\activations were 41–375 and 172–262, respectively. The two genomes showed significant differences. It was indicated that the two genomes responded differently to the WGD process. Non-additively expressed genes represented a small portion (5%–18%). The results of transgressive-upregulated GO enrichment was related to heterosis, while -downregulated genes were related to genetic changes, indicating that super-parent expression gene expression plays an important role in response to WGD, including field trait performance and maintenance of genomic balance. Besides, the genome-wide expression level dominance (ELD) was biased toward the A genome; the parental expression pattern of most genes showed a high degree (70.2%–73.4%) of conservation. Moreover, for genes with non-additive expression, BIA_A, BIA_C, high_parents, down_parents, transgressive-upregulation, transgressive-downregulation, and silencing\activations, the intersection between the eight plants was small, and the proportion of the set was large. This indicates that there was a large difference in gene expression between eight plants. And the expression differences of the six model genes are consistent with the cluster analysis of traits, indicating that differential expression of genes lays the molecular basis for trait separation. Furthermore, the differential expression of the waxy synthetic pathway and the flowering pathway-related gene in eight individuals could also be used to answer the field performance of the trait. This experiment lays a foundation for further exploration of the reasons for the separation of the newly synthesized *B. napus*.

## Figures and Tables

**Figure 1 genes-12-00082-f001:**
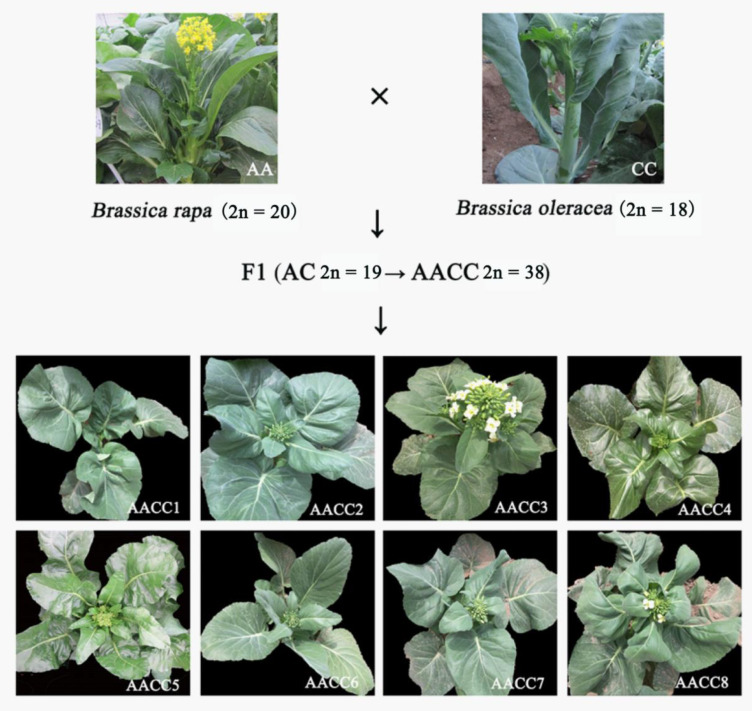
The materials used in the study. The picture shows female parent, male parent, eight second generation (F2) plants, and the process of obtaining the second generation (F2).

**Figure 2 genes-12-00082-f002:**
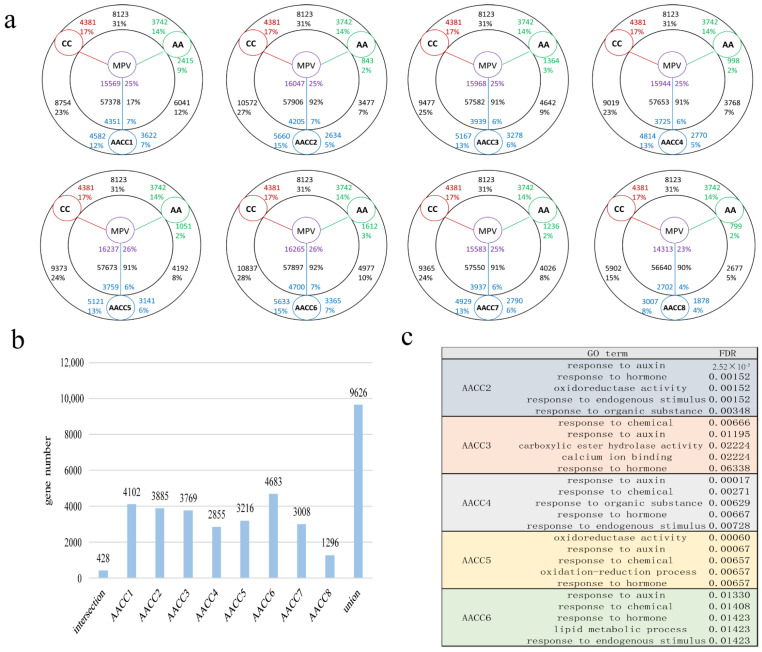
Differential, non-additive gene expressions in synthetic allopolyploids. (**a**) Differences in gene expression between the female parent (AA), male parent (CC), mid-parent expression values (MPV), and eight synthesis allotetraploids. (**b**) Expression of non-additive genes in eight single plants, intersection, and union. (**c**) Analysis of Gene Ontology (GO) enrichment of non-additive expression genes in five single plants (The remaining three single plants were not significantly enriched).

**Figure 3 genes-12-00082-f003:**
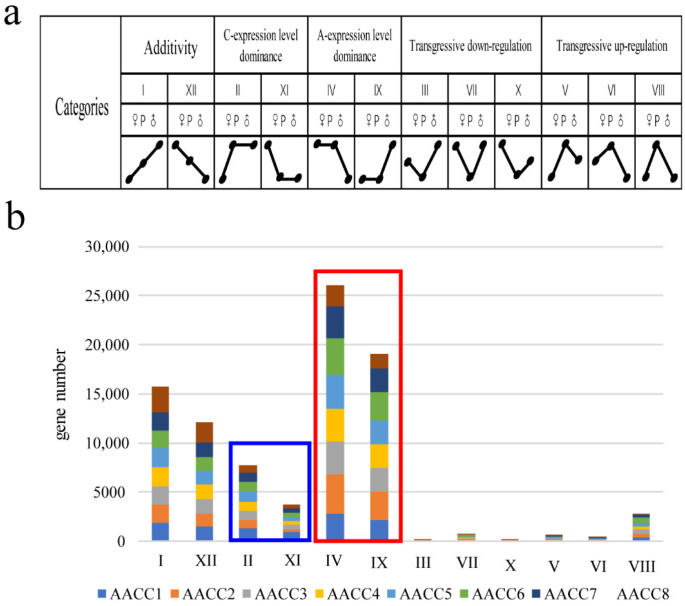
Expression level dominance in synthetic allopolyploids. (**a**) Expression patterns of 12 types of homologous genes; (**b**) Number of genes in each type of homologous gene in the eight allopolyploids.

**Figure 4 genes-12-00082-f004:**
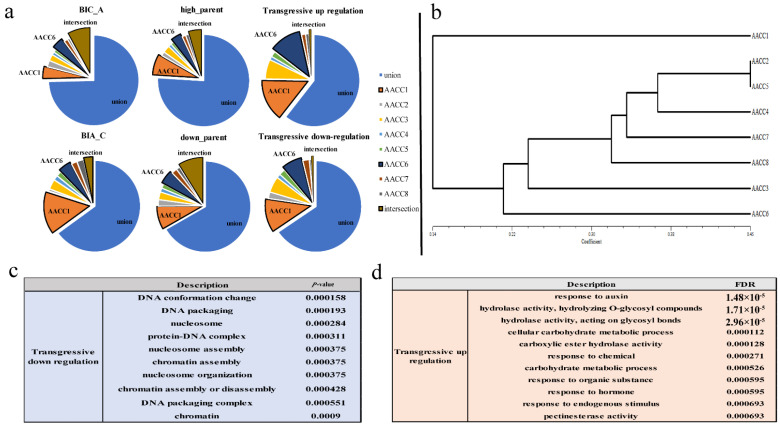
Analysis of transgressive expression and trait cluster. (**a**) Ratio of difference set, intersection, and set of six gene expression patterns in eight plants; (**b**) Graph of Character clustering; (**c**) Analysis of GO enrichment of transgressive-down regulation; (**d**) Analysis of GO enrichment of transgressive-up regulation.

**Figure 5 genes-12-00082-f005:**
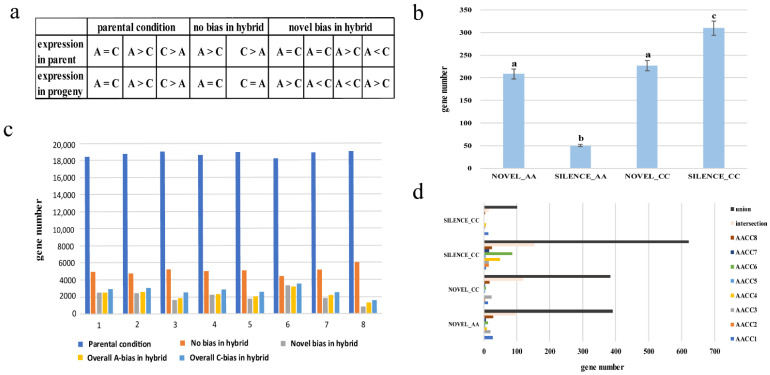
Homoeolog expression and novel expression/silencing analysis. (**a**) Gene classification between the parental diploids and synthetic allopolyploids; (**b**) the gene numbers of five patterns in eight plants; (**c**) Significant analysis of the difference in novel expression/silencing genes; (**d**) Difference set, intersection, and set of novel expression/silencing genes in eight plants.

**Figure 6 genes-12-00082-f006:**
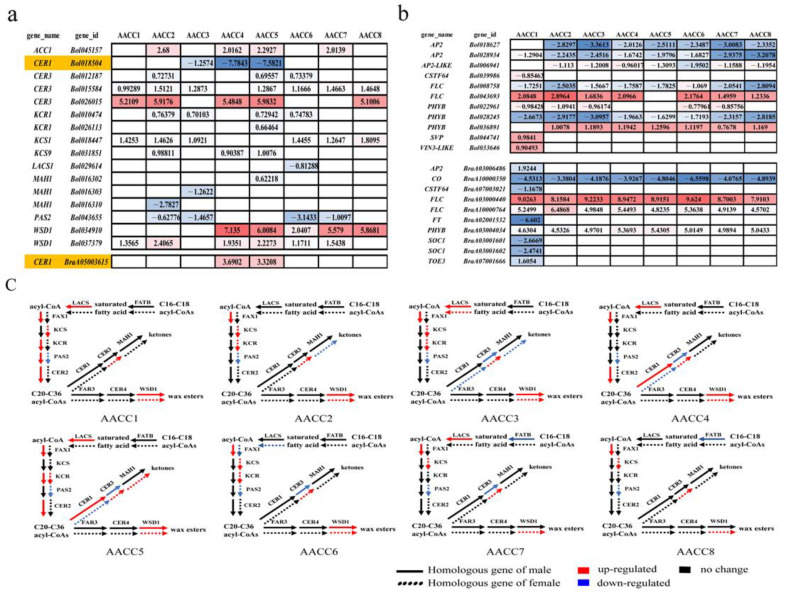
Analysis of the genes related to wax synthesis and flowering pathway. (**a**) Analysis of the genes related to wax synthesis (number is the value of log_2_ Fold Change); (**b**) Analysis of the genes related to the flowering pathway (the number is the value of log_2_ Fold Change); (**c**) Wax synthesis pathway showing routes for the production of wax. Gene names are color-coded to represent their different changes in expression levels in the F2 hybrid; the same color codes are used for the biosynthetic products to reflect their expected changes.

## Data Availability

The RNA data we sequenced was uploaded to the Big sub-database with the project number PRJCA004160.

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
