# Peer review of "Analysis of Transcriptional Changes in Different Brassica napus Synthetic Allopolyploids"

_genes, 2021, doi:10.3390/genes12010082_

Round 1

Reviewer 1 Report

Comments have been inserted into the text in the file attached.

The study is limited to synthetic allo-tetraploids with A cytoplasm. More information could be obtained with creation of a C cytoplasm allotetraploid for comparison, especially for genome bias in expression, and genome differences in gene silencing and other effects.

The study would draw more convincing conclusions with an F2 population at least 3 times larger, even up to 50.

There is clear evidence for homoelogous genetic exchange and segregation, in contrast to the stable meiosis in domestic B. napus. Possibly the haploid condition of the AC hybrid is conducive to homoelogous exchange.

English grammar needs improvement, various cases where a phrase is written as a stand alone sentence with the initial word capitalised.

Diagrams need to be clearly explained, as to what is shown and how it is shown. In some cases they are not helpful, even confusing, for this reason

Findings in the paper are not justified with such a small F2 population, such as for genome expression dominance, non-additive expressions, transgressive segregation etc. Such effects clearly occur, but the sample is too small for authoritative statements. The paper describes a preliminary study before a more thorough study.

Author Response

Thank the reviewers for comments and suggestions.The feedback was insightful and enabled us to improve the quality of our manuscript. We have provided point-by-point responses to reviewers’ comments, below.

Comment 1: The study is limited to synthetic allo-tetraploids with A cytoplasm. More information could be obtained with creation of a C cytoplasm allotetraploid for comparison, especially for genome bias in expression, and genome differences in gene silencing and other effects.

Response 1The reviewer’s suggestions are very meaningful. Thanks to the reviewers’ comments. The research team has created material of a C cytoplasm allotetraploid and will explore them further based on the reviewers’ comments.

Comment 2: The study would draw more convincing conclusions with an F2 population at least 3 times larger, even up to 50.

Response 2The reviewer’s suggestions are very meaningful. Thanks to the reviewers’ comments. However, considering the limitations of scientific research funding, the number of F2 groups has been reduced. With the popularization of sequencing, F2 population will be further expanded, and further exploration will be carried out according to suggestions of reviewers.

Comment 3: There is clear evidence for homoelogous genetic exchange and segregation, in contrast to the stable meiosis in domestic B. napus. Possibly the haploid condition of the AC hybrid is conducive to homoelogous exchange.

Response 3The reviewer’s suggestions are very meaningful. Thanks to the reviewers’ comments. The research team has AC haplotype materials, and we will further investigate this problem based on the comments of the reviewers.

Comment 4: English grammar needs improvement, various cases where a phrase is written as a stand alone sentence with the initial word capitalised.

Response 4Thanks to the reviewers’ comments. The content of the manuscript has been improved in language. For example:

“At the genetic level, loss of parental and/or appearance of novel sequences at the initial stage of allopolyploidization are common events.” was change into “ At the genetic level, loss of parental and/or appearance of novel sequences are common events at the initial stage of allopolyploidization.”

“We extend earlier findings of homoeolog expression bias and expression level dominance by parsing expression patterns in eight synthetic allopolyploids and their parents, as well as relationship between gene expression difference and trait separation in eight plants, as well as analysis of gene expression in waxy synthetic pathway and flowering pathway.” was changed into “We extend earlier findings of homoeolog expression bias and expression level dominance by parsing expression patterns in eight synthetic allopolyploids and their parents. The experiment also analyzed the relationship between gene expression difference and trait separation in eight plants, taking the waxy and flowering traits as an example to show.”

“Besides, GO enrichment analysis gene expression of transgressive-up regulation and -down regulation found that the transgressive-up regulation gene was mainly enriched in the process of stress resistance,” was changed into “Besides, gene expression of transgressive-up regulation and -down regulation were analyzed for GO enrichment.”

Comment 5: Diagrams need to be clearly explained, as to what is shown and how it is shown. In some cases they are not helpful, even confusing, for this reason

Response 5Thanks to the reviewers’ comments. The Diagrams of the manuscript has been revised according to the reviewers’ suggestions. For example:

Figure 1. The materials used in the study.” was changed into “Figure 1. The materials used in the study. The picture shows female parent, male parent, eight F2 plants, and the process of obtaining F2.”

Comment 6: Findings in the paper are not justified with such a small F2 population, such as for genome expression dominance, non-additive expressions, transgressive segregation etc. Such effects clearly occur, but the sample is too small for authoritative statements. The paper describes a preliminary study before a more thorough study.

Response 6The reviewer’s suggestions are very meaningful. Thanks to the reviewers’ comments. Our team will further expand the F2 population, select different hybridization combinations, and further study the genetic variation analysis of the offspring to obtain more convincing conclusions.

Reviewer 2 Report

The article is  interesting, there are no observations in the manuscript.

I know that the authors have worked hard to write this manuscript.

Line 95 Pleaseb show the parameters of read filtering

English grammar needs improvement, various cases where initial word capitalised e.g. line 23 "And" Line 26-27 Please change font for "...among the selfing offspring" Line 255 Please replace was with were 

Author Response

Thank the reviewers for comments and suggestions.The feedback was insightful and enabled us to improve the quality of our manuscript. We have provided point-by-point responses to reviewers’ comments, below.

Comment 1: The article is interesting, there are no observations in the manuscript. I know that the authors have worked hard to write this manuscript.

Response 1Thank you very much for reviewer’s affirmation and suggestions.

Comment 2: Line 95 Please show the parameters of read filtering

Response 2The data filtering criteria are: (1): Remove reads with adapter (adapter); (2) Remove N (N means that the base information cannot be determined) is greater than 10% of reads; (3): Remove low-quality reads (reads whose Qphred <= 20 bases account for more than 50% of the entire read length).

Comment 3: English grammar needs improvement, various cases where initial word capitalised e.g. line 23 "And" Line 26-27 Please change font for "...among the selfing offspring" Line 255 Please replace was with were 

Response 3: Thanks to the reviewers’ comments. The content has been revised according to the reviewers’ suggestions.

Round 2

Reviewer 1 Report

Different sections results and discussion not consistent and not drawn together.

Need total consistency referring to A genome and C genome results in the same respective order throughout. Current presentation is confusing comparing different sentences. Are up-regulation, down regulation, expression dominance and gene silencing consistently higher for the A or C genome in the allo-polyploid by a large or small amount, and does the pattern differ among the 8 F2 ? Silencing appears to be clearly greater for C genome.

No explanation why only 8 F2 available. The paper would be more convincing with >25 F2, even 50.

A Cytological observation of meiosis in the F2 plants could indicate chromosome pairing and chromosomal physical differences among the 8 as a result of homeologous pairing.

More clarity needed. More attention on additive expressions among the 8.

In literature review check for references to natural ploidy of B. napus ~ 1,000 AD.

Author Response

Thank the reviewer for comments and suggestions.The feedback was insightful and enabled us to improve the quality of our manuscript. We have provided point-by-point responses to reviewers’ comments, below.

Comment 1: Need total consistency referring to A genome and C genome results in the same respective order throughout. Current presentation is confusing comparing different sentences. Are up-regulation, down regulation, expression dominance and gene silencing consistently higher for the A or C genome in the allo-polyploid by a large or small amount, and does the pattern differ among the 8 F2 ? Silencing appears to be clearly greater for C genome.

Response 1: The reviewer’s suggestions are very meaningful. Thanks to the reviewers’ comments. With reference to the previous definition, there is an explanation for Expression level dominance. “In general, a higher number of transgressive upregulated (categories V, VI, and VIII) genes in the eight synthetic allopolyploids was observed relative to the number of downregulated (categories III, VII, and X) genes in the synthetic allopolyploids (14.8% vs. 3.9%). We examined all four progenies (II IV IX XI) for evidence of expression level dominance. The expression levels of genes belonging to categories IV, and IX were statistically equivalent to that of the A-genome parent, and genes of categories II and XI were the same as that of the C-genome parent.” According to the results of data analysis, an average of 16.3% more gene pairs [2,819 (21.8%) vs. 715 (5.5%)] exhibited expression level dominance toward the A-parent than the C-parent.

The data of up-regulation, down regulation, gene silencing cannot be used for the analysis of toward A-genome and C-genome. For the data of up-regulation and down regulation, we we performed GO enrichment analysis, the result showed that the transgressive-up regulation gene may play a role in heterosis, and transgressive-down regulation gene may explain genetic changes caused by the combination of two genomes. For the data of gene silencing and novel expression, the result showed that the CC genome showed a bigger change than AA genome. “CC genome showed a bigger change than AA genome” is different from toward A-genome and C-genome.

The pattern is the same among the 8 F2 plants.

Comment 2: No explanation why only 8 F2 available. The paper would be more convincing with >25 F2, even 50.

Response 2: The reviewer’s suggestions are very meaningful. Thanks to the reviewers’ comments. In the previous experiment, we made 3 cross combinations, but there are only 8 plants in F2 of this combination, and the selection of this combination is mainly due to differences in traits.

Comment 3: A Cytological observation of meiosis in the F2 plants could indicate chromosome pairing and chromosomal physical differences among the 8 as a result of homeologous pairing.

Response 3: The reviewer’s suggestions are very meaningful. Thanks to the reviewers’ comments. 

These 8 F2 plants are from the same F1 plant, and F1 is a tissue culture plant. At that time, flow cytometry experiments were performed on F1 and 8 F2, and the experiment showed that F1 and 8 F2 were allotetraploid. These data are all present in my doctoral Dissertation. When further exploring the F2 progeny, we will conduct cytological observations based on the comments of the reviewers.

Comment 4: More clarity needed. More attention on additive expressions among the 8.

Response 4: The reviewer’s suggestions are very meaningful. Thanks to the reviewers’ comments. 

Because most genes in F2 were additive expression. This model conforms to people's subjective ideas. So we only analyzed non-additive expression and it showed that these specifically expressed genes play an important role in field difference performance. 

Comment 5: In literature review check for references to natural ploidy of B. napus ~ 1,000 AD.

Response 5: The reviewer’s suggestions are very meaningful. Thanks to the reviewers’ comments. B. napus is an important oil crop and is a hybrid that was generated from a cross between B. rapa and B. oleracea approximately ~7500 years ago20. was changed to B. napus is an important oil crop and is a hybrid that was generated from a cross between B. rapa and B. oleracea approximately between 6800 and 12500 years ago20. based the article of Hong An et al. (An, H., et al. Transcriptome and organellar sequencing highlights the complex origin and diversification of allotetraploid Brassica napu. Nat commun 10, 2878(2019))